# Development of the life change adaptation scale for family caregivers of individuals with acquired brain injury

Yuka Shindo [*], Etsuko Tadaka

Department of Community Health Nursing, Graduate School of Medicine, Yokohama City University, Yokohama, Kanagawa, Japan

☯ These authors contributed equally to this work.

* t206705c@yokohama-cu.ac.jp

**Data Availability Statement:** All relevant data are within the paper and its Supporting Information files.

**Funding:** The authors received no specific funding for this work.

## Abstract

### Aim

Life changes due to the sudden onset of acquired brain injury (ABI) are drastic personal and social changes that require adaptation and are also an important indicator of the quality of life of family caregivers. However, there are no instruments for evaluating life change adaptation among family caregivers of individuals with acquired brain injury. This study aimed to develop the Life Change Adaptation Scale (LCAS) for family caregivers of individuals with ABI and examine its reliability and validity.

### Methods

A cross-sectional study was conducted using a self-reported questionnaire. A total of 1622 family caregivers of individuals with ABI who belonged to 82 associations for families of individuals with ABI were selected as eligible participants. The construct validity was evaluated using a confirmatory factor analysis. Internal consistency was calculated using Cronbach's alpha. The K6 was also administered to assess the criterion-related validity of the LCAS.

### Results

In total, 339 valid responses were received. The confirmatory factor analysis identified eight items from two domains, "Changes in the appraisal of caregiving resources" and "Changes in the health belief as a caregiver" (goodness of fit index = 0.963, adjusted goodness of fit index = 0.926, comparative fit index = 0.986, root mean square error of approximation = 0.043.) Cronbach's alpha was 0.84. The LCAS was negatively correlated with the K6 (r = -0.504; P<0.001).

### Conclusions

The LCAS is a brief, easy-to-administer instrument that is reliable and valid for family caregivers of individuals with ABI. This study contributes to the assessment and identification by family caregivers of individuals with ABI who require aid in adapting to life changes. Further

**Competing interests:** The authors have declared that no competing interests exist.

research should be undertaken to verify the predictive value in a longitudinal study and to attempt to apply the LCAS to assess a broader range of subjects in a wider range of settings.

## Introduction

Acquired brain injury (ABI), is defined as damage to the brain that occurs after birth and that is not related to a congenital disorder or degenerative disease [1, 2]. In Japan, ABI cases with cognitive and behavioral dysfunction are defined as "Higher brain dysfunction" [3]. ABI is an unforeseen condition with physical, cognitive and psychosocial deficits that significantly impact upon a person's ability to live a productive life, including increased aggression, poor memory, concentration difficulties and speech impairments [2, 4, 5]. ABI is most commonly attributed to traumatic brain injury (TBI) or stroke. In 2016, over 27.1 million new cases of TBI were reported worldwide, representing a 3.6% increase from 1990 [6]. In the same year, almost 69 million individuals worldwide were estimated to suffer from residual disabilities due to TBI [6]. In 2016, over 13.7 million new cases of stroke were reported worldwide, representing an 8.1% decrease from 1990 [7]. The incidence of stroke rapidly increases with age, doubling for each decade after age 55 [7, 8]. In the same year, almost 116.4 million individuals worldwide were estimated to suffer from residual disabilities due to stroke [7]. The overall burden of ABI in terms of the absolute number of people affected by ABI or who remained disabled due to ABI has increased across the globe [9].

The families of individuals with ABI play a substantial role in their support after acute hospitalization [10]. In Japan, approximately 90% of individuals with ABI are living with and require daily support from their family [3]. Thus, families of individuals with ABI assume the responsibility of long-term caregiver. Accumulated evidence indicates that the family caregivers of individuals with ABI experience negative health impacts, such as a high level of burden, anxiety, depressive symptoms and poor mental health and quality of life in association with the long length of time spent providing care [11–15].

There is international recognition of the need for psychosocial interventions to improve the quality of life of family caregivers. Several instruments measuring the outcomes of family caregivers were developed to identify family caregivers who are in need of intervention and were used to assess interventions [16]. However, there is substantial diversity in the outcomes, even after ABI. The main focus of previous studies and measures for family caregivers concerning individuals with ABI has been to identify the negative outcomes of the onset of ABI [17]. However, the family caregivers of individuals with ABI can experience not only negative adaptations to life changes but also positive adaptations following the onset of ABI, and such positive life change adaptations are just one of the benefits these individuals can achieve. Measuring the life change adaptations makes it possible to evaluate family caregivers of individuals with ABI and the health services offered by health professionals. It can also improve the quality of life for both the family caregivers of individuals with ABI and other members of their family, even if the individual with ABI has a severe disability.

In 1999, Bakas and Champion (1999) constructed the Bakas caregiving outcomes scale (BCOS) for family caregivers of individuals with stroke to assess life change adaptation. Life change when assuming the caregiving role for individuals with stroke is considered to be an adaptational outcome based on Lazarus' (1991) definitions of social functioning, subjective well-being, and somatic health [18, 19]. Bakas and Champion (1999) found life change adaptation in the family caregivers of individuals with stroke to be negatively related to mental health

and positively related to well-being [18]. However, the BCOS cannot be directly applied to ABI. While most strokes occur in older age (≥65 years) [20], most ABIs occur in adult age (16-44years) [21]. While the family caregivers of individuals with stroke are often in older age rather than middle-aged individuals (45-64years) [22], the family caregivers of individuals with ABI are often middle-aged or older. Developmental tasks differ between middle-aged and older individuals. In comparison to older individuals, middle-aged individuals are more frequently confronted with the roles of family control responsibility and the social responsibility, such as marriage, parenting and working to support the family. Thus, the life course and roles of the family caregiver differ between stroke and ABI. Furthermore, family caregivers of adult ABI patients are more likely than those of pediatric ABI patients to have their adaptational outcomes affected [23, 24]. Pediatric ABI patients complete their early childhood development with their disability [25, 26]. In contrast, loss may be felt when dealing with adult ABI patients, as caregivers will be reminded of a time when the patient had normal cognitive and emotional development without disability. Therefore, measuring the life change adaptation of family caregivers of individuals with ABI requires a longitudinal viewpoint with consideration of the long-term adaptational processes after adult ABI.

Atchley (1999) showed that middle-aged and older adults adapt to changes by using strategies to maintain continuity in their lives, both external and internal in the Continuity Theory [27, 28]. Continuity focuses on global external frameworks, including lifestyles, networks of social relationships, and activity profiles, and internal mental frameworks, including the persistence of a psychic structure of ideas, temperament, affect, experiences, preferences, dispositions, and skills, personal goals, or belief systems [28, 29]. The extent of continuity is determined by a here-and-now assessment made by the individual based on their self-remembered past [27]. Therefore, the life change adaptation of family caregivers of individuals with ABI can be explained with the Continuity Theory. The key to improving life change adaptation involves the specific new circumstances in family caregivers of individuals with ABI, the external dimension and the internal dimension. Thus, it is important to develop a new conceptual instrument, "The Life Change Adaptation Scale (LCAS)," for family caregivers of individuals with ABI that considers the life change adaptation a family caregiver experiences—which consist of external dimensions and internal dimensions—to measure, understand, explain and predict life change adaptation and facilitate intervention to help family caregivers of individuals with ABI.

The object of this research is to develop the Life Change Adaptation Scale (LCAS) for family caregivers of individuals with ABI by measuring adaptation to life changes caused by ABI, and to examine its reliability and validity. In this article, "individuals with acquired brain injury" refers to "disabled individuals with cognitive and behavioral dysfunction because of damage to the brain that occurs after birth and that is not related to a congenital disorder or a degenerative disease." "Family caregiver of individuals with ABI" refers to "a family relative that cares for or assists an individual with ABI in their daily lives." "Life change adaptation" refers to "the outcome of adaptation to changes in the appraisal of caregiving resources / the health belief of life of family caregivers due to acquired brain injury."

## Methods

### Phase 1: Developing the instrument

A first draft of the LCAS was initially developed following a critical review of the relevant literature. Articles were identified in PubMed, Ichushi-Web and PsycInfo according to the theme "life change" in family caregivers of individuals with ABI. The following search terms were used: "caregivers", "brain injury", "stroke", "encephalitis", "brain tumor", and "hypoxic

ischemic encephalopathy" and along with the MeSH terms, "adult" and "middle aged". These searches yielded nine articles [18, 19, 30–34]. The inclusion of an article was based on two criteria: 1) the article was related to the research experiences of family caregivers of individuals with ABI; 2) the article was associated with existing life change scales. Based on the literature review and the researchers' experiences, a first draft of the LCAS was constructed. This first draft contained 22 items.

To ensure the validity of the content of the first draft of the LCAS, ten experts were invited to rate the relevance of each item with regard to its content in relation to life changes in family caregivers of individuals with ABI. The experts were selected from experienced researchers, clinical support staff and family caregivers (two professors, two clinical psychologists, one social worker, and five family caregivers of individuals with ABI who belonged to an association for family caregivers of individuals with ABI). The responses were scored as follows: 1 = completely important, 2 = slightly important, 3 = slightly unimportant, 4 = not important at all. In addition, "unknown" was included to understand the difficulty level of an item. The author revised the wording of each item based on expert opinions (e.g., to avoid double-barreled questions and ambiguous wording). Consequently, the initial LCAS was refined to include fifteen items.

## Phase 2: Validating the instrument

**Participants and settings.** The survey was conducted among 1622 family caregivers of individuals with ABI who belonged to associations for families in Japan and were selected as eligible participants between September 2019 and November 2020. The 82 associations were selected from a list that is publicly available in Japan. Before sending the survey questionnaires, the author sent informed consent letters to the administrators of all associations of family caregivers of individuals with ABI. Thirty-nine associations (47.5%) consented to participate in the study. The survey questionnaires were distributed to each participant by the staff of each association. The study's inclusion criteria as follows: 1) caring for an individual with ABI, 2) ≥20 years old, 3) a family member of an individual with ABI, and 4) the individual with ABI developed ABI at 16–64 years old. The following exclusion criteria also had to be met: 1) not caring for an individual with ABI, 2) ≤19 years old, 3) not a family member of individual with ABI, 4) the individual with ABI developed ABI at <16 or >64 years old. The reasons for including "the individual with ABI developed ABI at 16–64 years old" and excluding cases in which ABI occurred in an individual at <16 or >64 years old are that adult ABI is not an injury in a stage that progresses with age, like pediatric ABI, and it is not an injury in a stage that declines with age, like ABI in old age, and it is in a stage that is associated with a consistent cognitive and psychosocial function. As a result, the adaptation process and outcomes for ABI are different. Data were collected between September 2019 and January 2020.

**Measures.** The participants' demographic characteristics included age, sex, and relationship to an individual with ABI. The individuals with ABIs' demographic characteristics included age, sex, age at the time of ABI occurrence, cause of ABI, period after ABI, and impairment.

The LCAS was scored on a seven-point Likert scale ranging from -3 (the most deteriorated), to 0 (did not change), to +3 (the most improved).

To assess the concurrent validity of the scale, the participants were also asked to assess each item in the Japanese version of the K6 [35]. This scale consists of six items measuring mental health on a 5-point Likert scale. Each question rated the frequency of distress symptoms. The responses were scored as follows: 0 = none of the time, 1 = a little of the time, 2 = some of the time, 3 = a lot of the time, 4 = all of the time; thus, the total score ranges from 0 to 24. High

scores on the K6 indicate a low level of mental health. In this scale, Cronbach's alpha coefficient was 0.88. The area under the receiver operating characteristic curve of this scale indicated excellent screening ability for DSM-IV (Diagnostic and Statistical Manual of Mental Disorders, 4th edition) mood and anxiety disorders [35]. A previous study considered the K6 to be a continuous scale in which higher scores reflect lower levels of mental health [36]. In addition, another study explained that there was a correlation between an adverse impact on caregivers' mental health and the negative perception regarding life changes [5]. Based on the findings from previous studies, we predicted that mental health deterioration in cases with higher K6 scores reflected negative life change adaptation with lower LCAS scores. Thus, the author used this scale as an indicator of concurrent validity.

**Statistical analyses.** IBM SPSS ver. 22.0 and Amos 24.0 (SPSS Inc., Chicago, Illinois, USA) were used to perform all statistical analyses.

An item analysis and exploratory factor analysis were conducted to evaluate the reliability and convergent validity of the initial LCAS. The criteria for the item analysis included rate of response difficulty (non-respondents:≥5%), distribution (one answer specific in a seven-point Likert scale: ≥85%), skewness and kurtosis (absolute values of <1.0 each), correlations of each item (correlation coefficient >0.6), a good-poor analysis (no significant differences between the highest scoring and lowest scoring groups), an item-total analysis (correlation coefficient between the item and the total score without that item ≥0.5).

After the item analysis, the total sample was randomly divided into two samples for cross-validation: in group 1, an exploratory factor analysis was performed; and in group 2, a confirmatory factor analysis was performed.

To assess the dimensionality of the LCAS, an exploratory factor analysis (maximum likelihood solution method) with promax rotation was performed on the development sample. Dimensionality was assessed based on eigenvalue >1.0, and a scree plot. Item loading needed to exceed 0.40. A confirmatory factor analysis (CFA) was then conducted to verify the construct validity. The goodness of fit index (GFI), adjusted goodness of fit index (AGFI), comparative fit index (CFI), and root mean square error of approximation (RMSEA) were used to evaluate the data model fit. The model was accepted if the GFI, AGFI, and CFI indices were ≥0.900 and the RMSEA was ≤0.050. Furthermore, criterion-related validity was examined using the K6 total score.

Internal consistency reliability was evaluated by calculating Cronbach's alpha coefficient for the LCAS, with alpha ≥ 0.70 considered acceptable.

**Ethical considerations.** The Institutional Review Board of the Medical Department of Yokohama City University School approved this study on July 17th, 2019 (No. A190700007). All study participants provided their written informed consent and completed the questionnaire, which was unsigned to ensure the anonymity of the participants. The informed consent form explained the voluntary nature of participation, management of data, and intention to publish the results.

## Results

### Demographic characteristics

In total, 398 individuals returned the questionnaire and 339 (85.1%) individuals met the criteria for inclusion. A total of 59 families were excluded because the individual with ABI developed ABI at <16 years old (n = 37) or ≥65 years old (n = 8), there was no response (n = 8) or there was no response to the LCAS (n = 6). Tables 1 and 2 shows demographic characteristics. Family caregivers of individuals with ABI ranged in age from 31 to 83 years, with an average age of 63.0 (SD = 10.1). A total of 75.5% of family caregivers were female. Individuals with ABI

**Table 1. Demographic characteristics of family caregivers.**

| | | n = 339 | |
| --- | --- | --- | --- |
| | | Number or Mean±SD[a] | % or (Range) |
| Age (n = 325) | | 63.0±10.1 | (31–83) |
| | <25 | 0 | 0.0 |
| | 25–34 | 2 | 0.6 |
| | 35–44 | 10 | 2.9 |
| | 45–54 | 58 | 17.1 |
| | 55–64 | 103 | 30.4 |
| | 65–74 | 107 | 31.6 |
| | 75< | 45 | 13.3 |
| | Missing | 14 | 4.1 |
| Age at the time of ABI (n = 320) | | 51.1±10.0 | (19–76) |
| | 16–18 | 0 | 0.0 |
| | 19–24 | 3 | 0.9 |
| | 25–34 | 12 | 3.5 |
| | 35–44 | 54 | 15.9 |
| | 45–54 | 143 | 42.2 |
| | 55–64 | 78 | 23.0 |
| | 65< | 30 | 8.8 |
| | Missing | 19 | 5.6 |
| Sex (n = 332) | Female | 256 | 75.5 |
| | Male | 76 | 22.4 |
| | Missing | 7 | 2.1 |
| Relationship to individual with ABI (n = 335) | Parent | 178 | 52.5 |
| | Spouse | 142 | 41.9 |
| | Son/daughter | 1 | 0.3 |
| | Sibling | 13 | 3.8 |
| | Other | 1 | 0.3 |
| | Missing | 4 | 1.2 |

[a]SD: standard deviation

ranged in age from 21 to 77 years, with an average age of 49.5 (SD = 12.1). A total of 81.1% of individuals with ABI were male. The length of care ranged from 0.5 to 40 years, with an average length of care of 12.4 years (SD = 8.0).

## Item analysis

Table 3 shows the item analysis results. Item 8 was excluded according to the item-total analysis. The correlation coefficient between item 2 and 9 was higher than 0.6. Item 2 was excluded according to the item difficulty. Thus, an exploratory factor analysis with promax rotation was performed for the thirteen remaining items.

## Factor structure

Table 4 shows the factor loading for the exploratory factor analysis. The development model yielded eight items in two factors with eigenvalues and a scree plot. In our interpretation, factor I included four items (5, 6, 11, 14) interpretable as "Changes in the appraisal of caregiving resources" for family caregivers of individuals with ABI. Factor II included four items (1, 3, 4,

**Table 2. Demographic characteristics of individuals with ABI.**

| | | | n = 339 |
|---|---|---|---|
| | | Number or Mean±SD | % or (Range) |
| Age (n = 338) | | 49.5±12.1 | (21–77) |
| | <25 | 6 | 1.8 |
| | 25–34 | 26 | 7.7 |
| | 35–44 | 88 | 26.0 |
| | 45–54 | 97 | 28.6 |
| | 55–64 | 75 | 22.1 |
| | 65–74 | 43 | 12.7 |
| | 75< | 3 | 0.9 |
| | Missing | 1 | 0.3 |
| Age at the time of ABI (n = 339) | | 37.2±14.3 | (16–64) |
| | 16–18 | 32 | 9.4 |
| | 19–24 | 55 | 16.2 |
| | 25–34 | 69 | 20.4 |
| | 35–44 | 62 | 18.3 |
| | 45–54 | 73 | 21.5 |
| | 55–64 | 48 | 14.2 |
| | 65< | 0 | 0.0 |
| | Missing | 0 | 0.0 |
| Sex (n = 338) | Female | 63 | 18.6 |
| | Male | 275 | 81.1 |
| | Missing | 1 | 0.3 |
| Period after ABI (years) (n = 338) | | 12.4±8.0 | (0.5–40) |
| | <1 | 2 | 0.6 |
| | 1–3 | 40 | 11.8 |
| | 4–6 | 55 | 16.2 |
| | 7–9 | 50 | 14.7 |
| | 10≦ | 191 | 56.3 |
| | Missing | 1 | 0.3 |
| Cause of ABI (n = 339) | Trauma to the head | 174 | 51.3 |
| | Stroke | 144 | 42.5 |
| | Tumor | 13 | 3.8 |
| | Anoxia | 17 | 5.0 |
| | Infection | 15 | 4.4 |
| | Other | 1 | 0.3 |
| Impairment (n = 338) | Attention | 303 | 89.6 |
| | Problem solving | 303 | 89.6 |
| | Memory | 278 | 82.5 |
| | Language | 139 | 41.1 |
| | Spatial awareness | 81 | 24.0 |
| | Physical awareness | 40 | 11.9 |
| | Disease awareness | 39 | 11.6 |
| | Apraxia | 93 | 27.6 |
| | Topographical | 130 | 38.6 |
| | Behavioral control | 202 | 59.9 |

[a]SD: standard deviation

**Table 3. Item analysis of "the life change adaptation scale".**

|  |  |  |  |  |  |  |  |  |  | | | | |
|---|---|---|---|---|---|---|---|---|---|---|---|---|---|
|  |  |  | | | | | | | | | | | n = 339 |
| No. | Item | Item difficulty [a] | Population distribution [b] | | | | | | | Kurtosis / Skewness [c] | | Correlation of item [d] | Good-Poor analysis [e] | Item-Total correlation [f] |
|  |  |  | -3 | -2 | -1 | 0 | +1 | +2 | +3 | | | | | |
| 1 | Your attitude of not trying too hard alone on any issue | 3.5 | 17.4 | 23.0 | 21.5 | 22.1 | 7.1 | 5.0 | .3 | .379 | -.553 | – | .000** | .508** |
| 2 | Your feeling of respect for the person with acquired brain injury | 1.2 | 5.9 | 13.6 | 17.1 | 28.9 | 17.4 | 12.1 | 3.8 | -.021 | -.587 | + | .000** | .559** |
| 3 | Your relaxed mind that allows enjoyment of leisure activities and hobbies | 0.3 | 25.4 | 28.9 | 24.2 | 11.5 | 6.2 | 2.7 | .9 | .847 | .347 |  | .000** | .574** |
| 4 | Your mindset of respecting your own health | 0.3 | 9.1 | 15.0 | 18.0 | 19.2 | 20.9 | 14.2 | 3.2 | -.073 | -.937 | – | .000** | .579** |
| 5 | Your attitude of being considerate of the circumstances and feelings of others | 0.6 | 2.9 | 4.4 | 12.4 | 23.9 | 31.0 | 18.6 | 6.2 | -.460 | .038 | – | .000** | .573** |
| 6 | Your view toward social systems related to health and life for individuals with acquired brain injury | 2.9 | 3.2 | 6.8 | 13.6 | 24.5 | 29.8 | 16.2 | 2.9 | -.454 | -.160 | – | .000** | .671** |
| 7 | Your unique approach to relieving stress | 0.3 | 11.8 | 16.8 | 28.9 | 20.1 | 15.0 | 5.9 | 1.2 | .209 | -.549 | – | .000** | .705** |
| 8 | Your general impression of people with disabilities | 0.6 | 1.5 | 2.1 | 4.4 | 24.2 | 28.6 | 24.5 | 14.5 | -.521 | .385 | – | .000** | .434** |
| 9 | The relationship between you and the person with acquired brain injury | 0.9 | 9.7 | 10.9 | 17.1 | 33.6 | 13.6 | 10.3 | 3.8 | -.024 | -.444 | + | .000** | .558** |
| 10 | The relationship between you and family members other than the person with acquired brain injury | 0.6 | 8.0 | 11.2 | 18.9 | 37.8 | 13.3 | 7.7 | 2.7 | -.024 | -.137 | – | .000** | .530** |
| 11 | Your attitude of seeking help when needed | 0.0 | 3.8 | 6.5 | 11.8 | 27.4 | 33.6 | 12.1 | 4.7 | -.495 | .227 | – | .000** | .594** |
| 12 | Your sense of pride in yourself | 0.9 | 7.4 | 6.2 | 15.6 | 46.6 | 13.6 | 7.4 | 2.4 | -.276 | .469 | – | .000** | .664** |
| 13 | Interactions between you and your friends or acquaintances (except for family) | 0.9 | 13.0 | 15.9 | 24.5 | 28.6 | 10.6 | 5.9 | 0.6 | .076 | -.515 | – | .000** | .629** |
| 14 | Your sense of responsibility as a member of family | 0.6 | 5.3 | 4.7 | 10.6 | 40.7 | 16.5 | 14.7 | 6.8 | -.265 | .128 | – | .000** | .614** |
| 15 | Your outlook on your life going forward | 0.9 | 22.1 | 24.5 | 27.7 | 14.5 | 6.5 | 3.8 | 0.0 | .554 | -.297 | – | .000** | .633** |

**: P<0.001

Exclusion criteria of the item analysis

a: The percentage of no answers was over 5% of the sample.

b: Item with a score (-3 to +3) of 85% or higher in the sample.

c: Absolute value of skewness or kurtosis was less than -1 or greater than 1.

d: Correlation was over 0.6.

e: Difference in the average score between the highest scoring group and the lowest scoring group is not a significant difference (P≥0.05).

f: The correlation coefficient between the item and the total of all the items (but with exception of the item) has a correlation coefficient of 0.5 or lower.

15) interpretable as "Changes in the health belief as a caregiver" for family caregivers of individuals with ABI. The factor loading was >0.40 for each factor. The cumulative contribution of the two factors explained 53.1% of the variance. The correlation coefficient between the two factors was 0.57.

Fig 1 shows the factor loading for the confirmatory factor analysis of the LCAS (Fig 1). The model fit showed GFI = 0.963; AGFI = 0.926; CFI = 0.986; RMSEA = 0.043, and nearly satisfied the appropriate criteria in each subject.

**Table 4. Exploratory factor analysis of "the life change adaptation scale".**

| No. | Item | Factor I | Factor II | Total scale communality |
|---|---|---|---|---|
| | | Changes in the appraisal of caregiving resources | Changes in the health belief as a caregiver | n = 170 |
| 6 | Your view toward social systems related to health and life for individuals with acquired brain injury | 0.840 | -0.020 | 0.687 |
| 5 | Your attitude of being considerate of the circumstances and feelings of others | 0.737 | -0.028 | 0.521 |
| 14 | Your sense of responsibility as a member of family | 0.631 | 0.048 | 0.436 |
| 11 | Your attitude of seeking help when needed | 0.570 | 0.092 | 0.393 |
| 3 | Your relaxed mind that allows enjoyment of leisure activities and hobbies | -0.175 | 0.976 | 0.788 |
| 15 | Your outlook on your life going forward | 0.149 | 0.624 | 0.517 |
| 1 | Your attitude of not trying too hard alone on any issue | 0.087 | 0.601 | 0.428 |
| 4 | Your mindset of respecting your own health | 0.188 | 0.564 | 0.475 |
| Cumulative contribution (%) | | 42.2 | 53.1 | |
| Factor correlation coefficients (r) | Factor I | 1.00 | 0.57 | |
| | Factor II | 0.57 | | |

Maximum likelihood solution method with promax rotation.

Missing data were excluded.

## Internal consistency and validity

Table 5 shows internal consistency and validity of the LCAS. Cronbach's alpha was 0.84 for the total scale and "Changes in the appraisal of caregiving resources" (Factor I) and "Changes in the health belief as a caregiver" (Factor II) were 0.79 and 0.79, respectively. A negative life change adaptation as well as a low LCAS score showed a significant amount of variance between mental health deterioration as well as a high K6 score (r = -0.50; P<0.001).

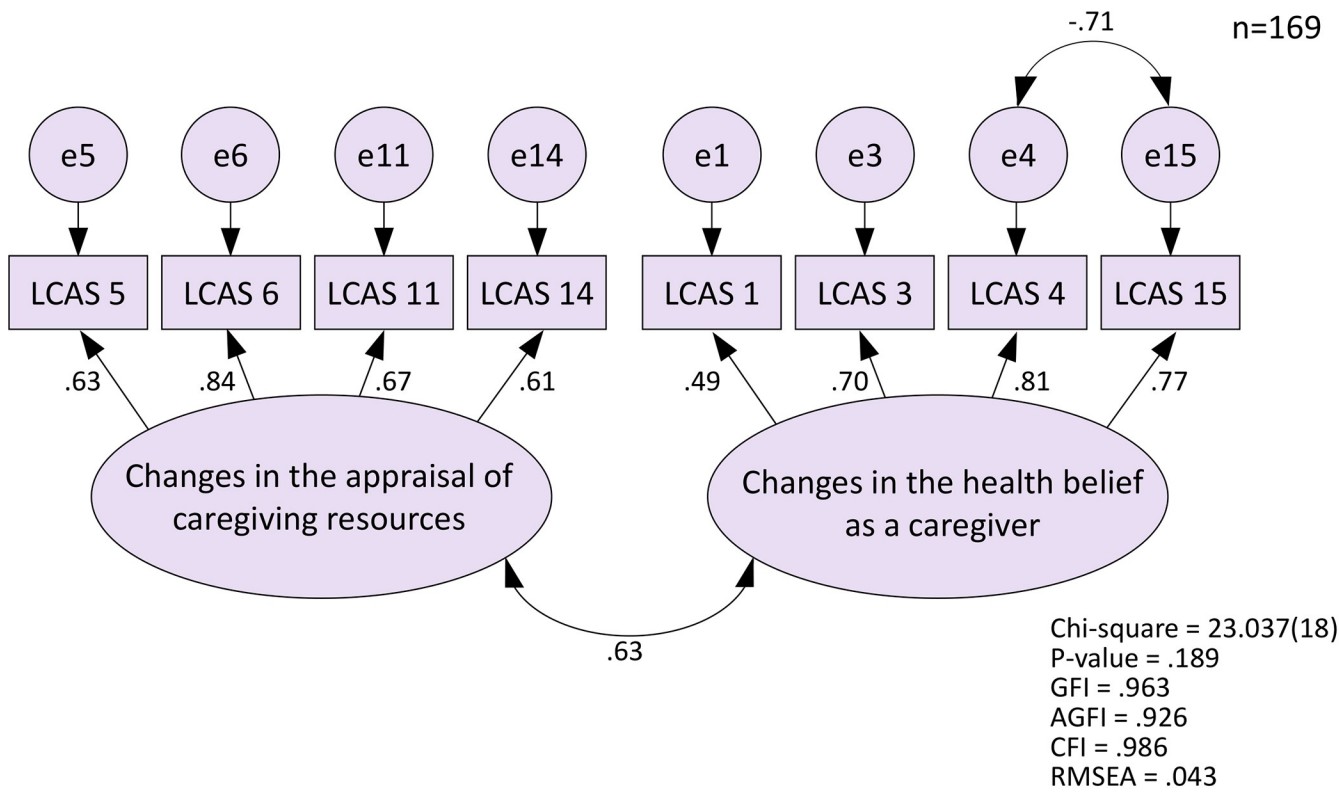

**Fig 1. Confirmatory factor analysis of "the life change adaptation scale".**

## Discussion

Life changes due to the sudden onset of acquired brain injury are drastic personal and social changes that require adaptation and are also an important indicator of the quality of life of family caregivers. However, there are no instruments for evaluating life change adaptation among family caregivers of individuals with acquired brain injury. To the best of our knowledge, the LCAS is the first scale developed for family caregivers of individuals with ABI, which measures life change adaptation caused by ABI. The LCAS demonstrated adequate reliability

**Table 5. Internal consistency and criterion-related validity of "the life change adaptation scale".**

| Factors | Mean (SD [a]) | The K6 [b] | Chronbach's alpha |
|---|---|---|---|
| I: Changes in the appraisal of caregiving resources | 1.7 (4.2) | -0.40* | 0.79 |
|  | n = 326 | n = 312 | n = 304 |
| II: Changes in the health belief as a caregiver | -4.0 (4.6) | -0.48* | 0.79 |
|  | n = 324 | n = 313 | n = 304 |
| Total 8 items | -2.2 (7.7) | -0.50* | 0.84 |
|  | n = 314 | n = 304 | n = 304 |

Pearson's correlation coefficients between the total score of validity measure of the LCAS

[a]SD: standard deviation

[b]The K6: Japanese version of the K6 (Furukawa et al., 2008)

*: P<0.001

(Cronbach's alpha was 0.84) and validity (r = 0.509; P<0.001, between the K6 scale). The dimensionality was confirmed by the CFA, which indicated a good fit (GFI = 0.963, AGFI = 0.926, CFI = 0.986, RMSEA = 0.043). The received response rate was low (20.9%). Furthermore, the demographic characteristics of non-responders were unknown, so the sample may have been biased. However, our response rate was similar to that of a previously published study polling the same population [37]. Regarding the demographics of the family caregivers of individuals with ABI, the caregivers were mostly women (75.5%). The average age at which the individual with ABI developed ABI was 37.2 years (SD = 14.3). According to the official evaluation by the Japanese government and a previous study, this is nearly identical to the profile of participants in the survey on the family caregivers of individuals with ABI [17]. Thus, the sample was deemed representative of the population of family caregivers of individuals with ABI.

The original point is to put forward the new model as opposed to the previous model proposed by Bakas [18, 19] with regard to three points: demonstration among not only the family caregivers of stroke patient, but for several types of diseases associated with ABI, focus on the individuals with ABI who developed ABI from 16–64 years of age and carry out recruitment not from hospitals but from the community. The majority of family members being 55–74 years old and the length of time since the injury being over 10 years were considered to have influenced the findings regarding adaptation. In general, families who have parents in this age range are more likely to have adult children who are no longer dependent. The family caregivers of individuals with ABI must fill the role of caring for an adult who is now dependent after having fulfilled their role of caring for their children. The family caregivers of individuals with ABI therefore need to fill a specific role that most family members never experience. This is likely to influence the adaptation of the family to the circumstances they face in relation to ABI.

The first factor of the LCAS, "Changes in the appraisal of caregiving resources," consists of four items: "Your attitude of being considerate to the circumstances and feelings of others", "Your view toward social systems related to health and life for individuals with acquired brain injury", "Your attitude of seeking help when needed", and "Your sense of responsibility as a family member." These items indicate the specificity of changes in familial and social resources which were given and received in order to carry out their daily living by the onset of ABI, such as responsibility for the family caregiver and interacting with social systems. Previous studies have described the changes in the roles to be endemic to family life after ABI [38, 39]. Another study also pointed out that the family adaptation to brain injury represents an effort to bring a new level of functioning to a family [40]. That is, "Changes in the appraisal of caregiving resources" of the LCAS has potential utility for evaluating the changes in new caregiving resources which are needed to adapt to the changing family roles and functions among the family caregivers of individuals with ABI.

The second factor of the LCAS, "Changes in the health belief as a caregiver", consists of four items: "Your attitude of not trying too hard alone on any issue", "Your relaxed mind that allows enjoyment of leisure activities and hobbies", "Your mindset of respecting your own health", and "Your outlook on your life going forward". These items indicate the specificity of changes in the belief which reflects perspective on their own health. There is strong evidence that the physical and emotional health of family caregivers of individuals with ABI are affected by the onset of ABI [13, 41–43]. Nonetheless, the strategies for managing their health still remain ambiguous. A previous study suggested that they lacked personal time for self-care even though they were aware that their own health was declining [44]. Although respite care allows caregivers to take some personal time and relax, it has been suggested that psychological characteristics are more important than respite care alone for the family caregiver of

individuals with ABI [43]. That is, "Changes in the health belief as a caregiver" of the LCAS has a potential utility for evaluating the changes in new health beliefs that are needed to adapt to the changing health situation among family caregivers of individuals with ABI.

## Limitation

The present study was associated with several limitations. First, a total of 75.5% of the family caregivers who participated in this survey, were female, while 81.1% of individuals with ABI were male. This does represent the traditional dynamics [17] but may contaminate assumptions about the meaning of the factors. However, this study design did not allow for a determination of causality between the LCAS response and the gender difference in family caregivers of individuals with ABI. Therefore, further longitudinal studies should be performed to verify the LCAS's predictive ability.

Second, the survey participants only included individuals who had joined associations of family caregivers of individuals with ABI, and the results may not be generalizable to the national population of family caregivers of individuals with ABI. Further studies should attempt to assess a broader range of subjects in a wider range of settings in order to validate these psychometric analyses.

## Supporting information

**S1 Appendix. The LCAS English version.**
(PDF)

**S2 Appendix. The LCAS Japanese version.**
(PDF)

## Acknowledgments

The author thanks Associate Professor Azusa Arimoto, Assistant Professors Kae Shiratani, Eriko Ito, and all members of the Department of Community Health Nursing, Graduate School of Medicine, Yokohama City University. Most of all, the author thanks all the family caregivers of individuals with ABI and the experts who graciously gave of their time and energy to participate in this study.

## Author Contributions

**Conceptualization:** Yuka Shindo, Etsuko Tadaka.

**Data curation:** Yuka Shindo.

**Formal analysis:** Yuka Shindo.

**Funding acquisition:** Etsuko Tadaka.

**Investigation:** Yuka Shindo.

**Methodology:** Yuka Shindo.

**Project administration:** Etsuko Tadaka.

**Resources:** Etsuko Tadaka.

**Software:** Yuka Shindo.

**Supervision:** Etsuko Tadaka.

**Validation:** Yuka Shindo, Etsuko Tadaka.

**Visualization:** Yuka Shindo.

**Writing – original draft:** Yuka Shindo.

**Writing – review & editing:** Yuka Shindo, Etsuko Tadaka.

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
