## [Decision Letter · Decision Letter 0]

24 Jul 2020

PONE-D-20-15809

Development of the Life Change Adaptation Scale for family caregivers of individuals with acquired brain injury

PLOS ONE

Dear Dr. Shindo,

Thank you for submitting your manuscript to PLOS ONE. After careful consideration, we feel that it has merit but does not fully meet PLOS ONE’s publication criteria as it currently stands. Therefore, we invite you to submit a revised version of the manuscript that addresses the points raised during the review process.

We look forward to receiving your revised manuscript.

Kind regards,

Manuel Fernández-Alcántara, Ph.D.

Academic Editor

PLOS ONE

Additional Editor Comments:

Dear authors,

Reviewer's have found you manuscript of being of great interest to the field. Please address the commentaries raised by reviewer 1 and also the major revision suggested by reviewer 2 (included in the doc flie attached to this review). We wait for a resubmission of your manuscript.

Journal Requirements:

2. In your Methods section, please provide additional information about the participant recruitment method and the demographic details of your participants. Please ensure you have provided sufficient details to replicate the analyses such as: a) the recruitment date range (month and year), b) a description of any inclusion/exclusion criteria that were applied to participant recruitment, c) a table of relevant demographic details, d) a statement as to whether your sample can be considered representative of a larger population, e) a description of how participants were recruited, and f) descriptions of where participants were recruited and where the research took place.

Reviewers' comments:

Reviewer's Responses to Questions

**Comments to the Author**

1. Is the manuscript technically sound, and do the data support the conclusions?

Reviewer #1: Yes

Reviewer #2: Partly

2. Has the statistical analysis been performed appropriately and rigorously? 

Reviewer #1: Yes

Reviewer #2: Yes

3. Have the authors made all data underlying the findings in their manuscript fully available?

Reviewer #1: Yes

Reviewer #2: No

4. Is the manuscript presented in an intelligible fashion and written in standard English?

Reviewer #1: No

Reviewer #2: Yes

5. Review Comments to the Author

Reviewer #1: This study aimed to develop the Life Change Adaptation Scale (LCAS) for family caregivers of individuals with ABI and to examine its reliability and validity. This reviewer feels that the authors contribute something important to the wellbeing of family caregivers for persons with ABI. With the average length of care delivery being 12 years, a family member’s mental and physical health is paramount. This article a good fit for the PLOS One Journal with some revisions and copy editing.

Those suggestions are as follows:

• The 4 point scale anchors described in the text (1 = completely important, 2 = slightly important, 3 = slightly unimportant, 4 =not important at all) are not the same as appear on the English-language version of the instrument (a 7 point scale from “The most deterioration” to “did not change” to “The most improved”) which is a substantially different instrument than the one described.

• On that note, the factor names for factor 1 does not seem to reflect the items (factor 1=“Changes in the living resources for caregiving” and factor 2=“Changes in the health belief as a caregiver”). Factor one includes ‘Your attitude of being considerate to the circumstances and feelings of others”, “Your view toward social systems related to health and life for individuals with acquired brain injury”, “Your attitude of seeking help when needed”, and “Your sense of responsibility as a family member.”—all of the items seem to assess attitudinal changes and not changes in resources—perhaps changes in APPRAISAL of resources?

• Throughout, there are some minor semantic updates required—all of which are likely attributable to the translation from Japanese.

• The methods overview in the abstract on page 2, line 28-29 doubles uses the word confirmed/confirmatory which adds redundancy. Likewise, in the conclusions overview on page 3, lines 41-42, the first sentence uses “family caregivers of individuals with ABI” twice.

• On page 3, lines 43 and 44, the sentence that reads “This study contributes toward assessing the life change adaptation and identifying family caregivers of individuals with ABI who require intervention.” could be stronger (e.g., this study contributes to the assessment and identification…”)

• On page 5, lines 82-83, the authors note that caregivers of stroke victims are often older age rather than middle age, and that ABI caretakers are often middle aged or older. This is important because the author references the ‘life course and roles of the family caregiver’. It may be useful to define what ‘middle aged’ and ‘older aged’ means in this context.

• On page 8, lines 138-140, there is another repeated use of ‘family caregivers of individuals with ABI’.

• On page 9, line 149, the authors refer to “type of disorder” but they likely mean impairment? In table 2 on page 14, the factors listed range from attention, to topographical, to physical awareness.

• On page 20, lines 286-287, there is a repeated use of the word strong/strongly. On the same page, line 289, the sentence that reads “lacked of personal time for self-care” could better be worded as “lacked personal time for self-care”.

• Page 20, the final sentence ending on line 295, is powerful and a great way to end the paper. The section on limitations that follows should precede that section to maximize impact.

• One other limitation to note is the overrepresentation of female respondents (76%) and male ABI patients (81%)—this does represent the traditional dynamics but may contaminate assumptions about the meaning of the factors.

• In PLOS ONE, the P value should be expressed as a capital P—that change can be made throughout.

Reviewer #2: Dear Authors,

This research will be of great interest to the ABI field. The main issue is that the ABI versus ageing process is not taken into consideration and this has implications for your analysis and the overshadows the likely benefits this measure could have within the ABI field. I would be most interested to read the revised version and encourage you to do this work to enhance the paper.

6. PLOS authors have the option to publish the peer review history of their article (what does this mean?). If published, this will include your full peer review and any attached files.

Reviewer #1: **Yes: **Kim A. Gorgens, Ph.D., ABPP & Hollis Lyman, MA

Reviewer #2: No

---

## [Author Response · Author response to Decision Letter 0]

28 Aug 2020

Thank you very much for your e-mail regarding our manuscript, “Development of the Life Change Adaptation Scale for family caregivers of individuals with acquired brain injury” (PONE -D-20-15809). We are grateful to know that it is potentially acceptable for publication in PLOS ONE. Please find attached a revised version of our manuscript.

Your comments and those of the reviewers were highly insightful and enabled us to greatly improve the quality of our manuscript. We include below our point-by-point responses to each of the comments of the reviewers as well as your own comments. 

We look forward to hearing from you regarding our re-submission. We would be happy to respond to any further questions and comments that you may have.

 

Response to Reviewers

To the comments of Reviewer #1

1. The 4 point scale anchors described in the text (1 = completely important, 2 = slightly important, 3 = slightly unimportant, 4 =not important at all) are not the same as appear on the English-language version of the instrument (a 7 point scale from “The most deterioration” to “did not change” to “The most improved”) which is a substantially different instrument than the one described.

Response: The 4-point scale was used to ensure the content validity of the first version of the LCAS by an expert check, and the 7-point scale was used to measure the life changes in the final version of the LCAS distributed to family caregivers of individuals with ABI. 

2. On that note, the factor names for factor 1 does not seem to reflect the items (factor 1=“Changes in the living resources for caregiving” and factor 2=“Changes in the health belief as a caregiver”). Factor one includes ‘Your attitude of being considerate to the circumstances and feelings of others”, “Your view toward social systems related to health and life for individuals with acquired brain injury”, “Your attitude of seeking help when needed”, and “Your sense of responsibility as a family member.”—all of the items seem to assess attitudinal changes and not changes in resources—perhaps changes in APPRAISAL of resources?

　

Response: As suggested, we have revised the factor names for factor 1 from “Changes in the living resources for caregiving” to “Changes in the appraisal of caregiving resources.”

3. Throughout, there are some minor semantic updates required—all of which are likely attributable to the translation from Japanese.

Response: As suggested, we have now had a professional medical editor whose native language is English proofread the revised manuscript (Japan Medical Communication, Inc.; https://www.japan-mc.co.jp).

The methods overview in the abstract on page 2, line 13-14 doubles uses the word confirmed/confirmatory which adds redundancy. Likewise, in the conclusions overview on page 3, lines 27-28, the first sentence uses “family caregivers of individuals with ABI” twice. 

Response: We have carefully corrected the highlighted portion as follows:

Line 13-14

“The construct validity was evaluated using a confirmatory factor analysis.”

Line 27-28

“The LCAS is a brief, easy-to-administer instrument that is reliable and valid for family caregivers of individuals with ABI.”

4. On page 3, lines 43 and 44, the sentence that reads “This study contributes toward assessing the life change adaptation and identifying family caregivers of individuals with ABI who require intervention.” could be stronger (e.g., this study contributes to the assessment and identification…”)

Response: As suggested, we have corrected line 28-29, as follows: 

“This study contributes to the assessment and identification by family caregivers of individuals with ABI who require aid in adapting to life changes.”

5. On page 5, lines 82-83, the authors note that caregivers of stroke victims are often older age rather than middle age, and that ABI caretakers are often middle aged or older. This is important because the author references the ‘life course and roles of the family caregiver’. It may be useful to define what ‘middle aged’ and ‘older aged’ means in this context.

　

Response: As suggested, we have added an explanation concerning what ‘middle aged’ and ‘older aged’ mean in this context (line 81-84).

“Developmental tasks differ between middle-aged and older individuals. In comparison to older individuals, middle-aged individuals are more frequently confronted with the roles of family control responsibility and the social responsibility, such as marriage, parenting and working to support the family.”

6. On page 8, lines 138-140, there is another repeated use of ‘family caregivers of individuals with ABI’. 

Response: As suggested, we corrected line 144-145.

“The survey was conducted with 1622 family caregivers of individuals with ABI who belonged to associations for families in Japan and were selected as eligible participants,”

7. On page 9, line 149, the authors refer to “type of disorder” but they likely mean impairment? In table 2 on page 14, the factors listed range from attention, to topographical, to physical awareness. 

Response: As suggested, we investigated impairment. We have now revised the text (line 167-168 and table 2).

8. On page 20, lines 286-287, there is a repeated use of the word strong/strongly. On the same page, line 289, the sentence that reads “lacked of personal time for self-care” could better be worded as “lacked personal time for self-care”. 

Response: As suggested, we have revised lines 334-336 and 336-338.

9. Page 20, the final sentence ending on line 295, is powerful and a great way to end the paper. The section on limitations that follows should precede that section to maximize impact. 

Response: We appreciate the reviewer’s suggestion. The section has been moved to a point after the limitations section (line 317-343).

10. One other limitation to note is the overrepresentation of female respondents (76%) and male ABI patients (81%)—this does represent the traditional dynamics but may contaminate assumptions about the meaning of the factors. 

Response: We appreciate the reviewer’s suggestion. The following has now been added to the limitations section (line 305-311):

“The present study was associated with several limitation. First, a total of 76.2% of the family caregivers who participated in this survey, were female, while 81.1% of individuals with ABI were male. This does represent the traditional dynamics [17] but may contaminate assumptions about the meaning of the factors. However, this study design did not allow for a determination of causality between the LCAS response and the gender difference in family caregivers of individuals with ABI. Therefore, further longitudinal studies should be performed to verify the LCAS’s predictive ability.”

11. In PLOS ONE, the P value should be expressed as a capital P—that change can be made throughout. 

Response: As suggested, we have corrected this point.

 

To the comments of Reviewer #2

Background

1. Information about the incidence of strokes across all age groups is not included. These need to be more clearly defined or the statements substantiated with more evidence. This is an important issue as the patients with stroke and their families have been excluded.

Response: The present study included patients with stroke and their families. Information on the incidence of stroke across all age groups is shown in references no.[7]. We have now added text on the incidence of stroke by age (lines 44-45).

“The incidence of stroke rapidly increases with age, doubling for each decade after age 55 [7,8]”

2. Given the argument for caregivers being within a defined age – was a pediatric ABI population considered? If not, then the nature of caring for an adult that was previously independent needs to be included in the background.

Response: As suggested, we have added the nature of caring for an adult that was previously independent needs to be included in the background (line 85-88). In addition, we have added the inclusion and exclusion criteria (line 152-156) along with the reasons for excluding pediatric ABI patients (line 156-162). 

Line 85-88

“Furthermore, family caregivers of adult ABI patients are more likely than those of pediatric ABI patients to have their adaptational outcomes affected [23,24]. Pediatric ABI patients complete their early childhood development with their disability [25,26]. In contrast, loss may be felt when dealing with adult ABI patients, as caregivers will be reminded of a time when the patient had normal cognitive and emotional development without disability.” 

Line 152-156

“inclusion criteria: 1) caring for an individual with ABI, 2) ≥20 years old, 3) a family member of an individual with ABI, and 4) the individual with ABI developed ABI at 16–64 years old. The following exclusion criteria also had to be met: 1) not caring for an individual with ABI, 2) ≤19 years old, 3) not a family member of individual with ABI, 4) the individual with ABI developed ABI at <16 or >64 years old.”

Line 156-162

“The reasons for including “the individual with ABI developed ABI at 16–64 years old” and excluding cases in which ABI occurred in an individual at <16 or ≥64 years old are that adult ABI is not an injury in a stage that progresses with age, like pediatric ABI, and it is not an injury in a stage that declines with age, like ABI in old age, and it is in a stage that is associated with a consistent cognitive and psychosocial function. As a result, the adaptation process and outcomes for ABI are different. 

3. What is the intention of developing this measure? What are the benefits for having such a measure for health services, health professionals and families? The rationale needs to be made clearer for what is missing and what the anticipated benefits.

Response: We appreciate the reviewer’s comments. The following has now been added to the revised manuscript (line 61-69):

“The main focus of previous studies and measures for family caregivers concerning individuals with ABI has been to identify the negative outcomes of the onset of ABI [17]. However, the family caregivers of individuals with ABI can experience not only negative adaptation to life changes but also positive adaptations following the onset of ABI, and such positive life change adaptations are just one of the benefits these individuals can achieved. This measure makes it possible to assess the life change adaptations in the family caregivers of individuals with ABI and of health services by health professionals. It can also improve the quality of life for both the family caregivers of individuals with ABI and other members of their family, even if the individual with ABI has a severe disability.”

Results

4. Why did some families receive the questionnaire that were not eligible? 

Response: We have now added an explanation concerning the number of people excluded (line 215-217).

“A total of 59 families were excluded because the individual with ABI developed ABI at <16 years old (n=37) or ≥65 years old (n=8), there was no response (n=8) or there was no response to the LCAS (n=6). “

5. Low response rate – please comment

Response: Because the number of received responses was low, the sample may have been biased. However, the sample was still representative. We have now mentioned this in the revised manuscript (line 290-293).

“The received response rate was low (20.9%). Furthermore, the demographic characteristics of non-responders were unknown, so the sample may have been biased. However, our response rate was similar to that of a previously published study polling the same population [36]. “

6. Co-morbidities? Given the age range of the person with ABI (up to 76 years), were other aspects of health/disability and care required taken into consideration? Difficult to determine how much of the family adjustment process is due in part to the ageing process for the person with the ABI and other family members.

Response: The LCAS was designed under the concept question of “How your life or attitude has changed since acquired brain injury struck your family?”, which can enable a self-assessment focused on ABI by the family. However, given the wide range in the potential age of the person with ABI, other aspects of health/disability and care required should also be taken into consideration. Whether and how ABI family caregivers will be affected by other aspects should be explored in a future study. 

Tables 3 and 4:

7. These tables are not legible and therefore the reviewer is unable to make comment. 

Response: We apologize for this error and have now revised the tables using the PLOS ONE style templates.

Discussion

8. The discussion needs to address the issues of ABI versus ageing. What was found between the ageing groups? Perhaps the analysis should be more about the comparison of groups by age or groups by time since injury

Response: As mentioned in our response to the sixth comment, we considered this study to be focused on adaptation to life changes for a wide range of ages of family caregivers, based on individuals with ABI who developed ABI from 16–64 years of age. Further studies should be performed to compare the patient groups by age or the length of time since the occurrence of ABI.

9. This measure has not addressed the changes related to the onset of an ABI – there are too many variables that have not been taken into account Or addressed in this section.

Response: As mentioned in our response to the sixth comment, we considered the LCAS to be able to extract only adaptations associated with ABI. The exact distinction between ABI, aging and other factors may be a subject for a future study. The following has now been added on this point to the revised manuscript (line 300-304):

“The original point is to put forward the new model as opposed to the previous model proposed by Bakas [18,19] with regard to three points: demonstration among not only the family caregivers of stroke patient, but for several types of diseases associated with ABI, focus on the individuals with ABI who developed ABI from 16–64 years of age and carry out recruitment not from hospitals but from the community.”

Editorial Requests:

Response: We appreciate the advice and have now used the PLOS ONE style templates to prepare our revised manuscript. 

2. In your Methods section, please provide additional information about the participant recruitment method and the demographic details of your participants. Please ensure you have provided sufficient details to replicate the analyses such as a) the recruitment date range (month and year), b) a description of any inclusion/exclusion criteria that were applied to participant recruitment, c) a table of relevant demographic details, d) a statement as to whether your sample can be considered representative of a larger population, e) a description of how participants were recruited, and f) descriptions of where participants were recruited and where the research took place.

Response: As suggested, we have now added the information requested. 

a) Line 144-146

“The survey was conducted among 1622 family caregivers of individuals with ABI who belonged to associations for families in Japan and were selected as eligible participants between September 2019 and November 2020.”

b) Line 150-156

“Of these, 39 associations (47.5%) participated in the study, and a questionnaire was distributed to the family caregivers of individuals with ABI who met the following inclusion criteria: 1) caring for an individual with ABI, 2) ≥20 years old, 3) a family member of an individual with ABI, and 4) the individual with ABI developed ABI at 16–64 years old. The following exclusion criteria also had to be met: 1) not caring for an individual with ABI, 2) ≤19 years old, 3) not a family member of individual with ABI, 4) the individual with ABI developed ABI at <16 or >64 years old.”

c) P.14-16

The relevant demographic and clinical characteristics of this study are shown in Table 1 (p.14-15) and Table 2 (p.15-16). We have now added the number of respondents to each variable.

d) Line 293-299

“Regarding the demographics of the family caregivers of individuals with ABI, the caregivers were mostly women (75.5%). The average age at which the individual with ABI developed ABI was 37.2 years (SD = 14.3). According to the official evaluation by the Japanese government and a previous study, this is nearly identical to the profile of participants in the survey on the family caregivers of individuals with ABI [17]. Thus, the sample was deemed representative of the population of family caregivers of individuals with ABI.”

e) and f) Line 146-148

“The 82 associations were selected from a list that is publicly available in Japan. The author sent informed consent letters and questionnaires to the administrators of each association. Each participant was asked to complete the voluntary self-administered anonymous questionnaire by the staff of each association.”

---

## [Decision Letter · Decision Letter 1]

23 Sep 2020

PONE-D-20-15809R1

Development of the life change adaptation scale for family caregivers of individuals with acquired brain injury

PLOS ONE

Dear Dr. Shindo,

Thank you for submitting your manuscript to PLOS ONE. After careful consideration, we feel that it has merit but does not fully meet PLOS ONE’s publication criteria as it currently stands. Therefore, we invite you to submit a revised version of the manuscript that addresses the points raised during the review process.

Reviewer's have suggested some minor revision to the manuscript before final acceptance. Please address Reviewer 1 modifications in the attached file and Reviewer 2 commentaries included at the end of this letter.

We look forward to receiving your revised manuscript.

Kind regards,

Manuel Fernández-Alcántara, Ph.D.

Academic Editor

PLOS ONE

Reviewers' comments:

Reviewer's Responses to Questions

**Comments to the Author**

1. If the authors have adequately addressed your comments raised in a previous round of review and you feel that this manuscript is now acceptable for publication, you may indicate that here to bypass the “Comments to the Author” section, enter your conflict of interest statement in the “Confidential to Editor” section, and submit your "Accept" recommendation.

Reviewer #1: All comments have been addressed

Reviewer #2: All comments have been addressed

2. Is the manuscript technically sound, and do the data support the conclusions?

Reviewer #1: Yes

Reviewer #2: Yes

3. Has the statistical analysis been performed appropriately and rigorously? 

Reviewer #1: Yes

Reviewer #2: I Don't Know

4. Have the authors made all data underlying the findings in their manuscript fully available?

Reviewer #1: Yes

Reviewer #2: No

5. Is the manuscript presented in an intelligible fashion and written in standard English?

Reviewer #1: No

Reviewer #2: Yes

6. Review Comments to the Author

Reviewer #1: There are a few remaining copy edits that would make the document stronger throughout--very minor edits.

Reviewer #2: The authors have addressed the reviewers previous comments. This paper is reading well. There are a few minor revisions needed:

1. Please re-write the sentences on lines 146-152 so that the procedure is clearer to the reader.

2. Table 1 – Change ‘relationship to ABI’ to ‘relationship to individual with ABI’ (so that the terminology is consistent with Table 2.

3. Table 3 is still not legible. Please amend.

4. Table 5 – align the factors to the left of the column

5. The discussion needs to address issues related to the majority of family members are aged 55-74 years and length of time since injury – and how this impacted on the findings regarding adaptation. This does impact on life issues that are impacting on the family. For instance, families who have parents in this age range are more likely to have adult children who are no longer dependent. This is likely to therefore impact on the adaptation of the family to the circumstances they face in relation to ABI.

6. Limitations also need to be included

7. PLOS authors have the option to publish the peer review history of their article (what does this mean?). If published, this will include your full peer review and any attached files.

Reviewer #1: **Yes: **Kim A. Gorgens, Ph.D., ABPP

Reviewer #2: No

---

## [Author Response · Author response to Decision Letter 1]

8 Oct 2020

Thank you very much for your e-mail regarding our manuscript, “Development of the life change adaptation scale for family caregivers of individuals with acquired brain injury” (PONE -D-20-15809). We are grateful to know that it is potentially acceptable for publication in PLOS ONE. Please find attached a revised version of our manuscript.

Your comments and those of the reviewers were highly insightful and enabled us to greatly improve the quality of our manuscript. We include below our point-by-point responses to each of the comments of the reviewers as well as your own comments. 

We look forward to hearing from you regarding our re-submission. We would be happy to respond to any further questions and comments that you may have.

Response to Reviewers

To the comments of Reviewer #1

1. Line 65 should read “these individuals can achieve.” On the same line through 66 it is unclear what “this measure” is referring to. 

Response: “This measure” refers to measuring the life change adaptations. We have revised the text as follows (line 63-65 and line 66-67):

Line 63-65

“However, the family caregivers of individuals with ABI can experience not only negative adaptations to life changes but also positive adaptations following the onset of ABI, and such positive life change adaptations are just one of the benefits these individuals can achieve.”

Line 66-67

“Measuring the life change adaptations makes it possible to evaluate family caregivers of individuals with ABI and the health services offered by health professionals.”

2. On line 122 PsychInfo is improperly referred to as “Psycho-Info”. 

Response: We have written it according to the American Psychological Association (https://www.apa.org/pubs/databases/psycinfo/) (line 122-123).

3. Line 130 is missing a period after “22 items”. 

Response: We have added a period after “22 items” (line 130).

4. Line 240 could benefit from copy editing (“The percentage of at least either one of response is higher than 85% of the sample.)

Response: We have corrected the text as follows (line 248):

Line 248

“Item with a score (-3 to +3) of 85% or higher in the sample.” 

5. The clarification of how middle-age and older-age differ in developmental tasks in lines 81-84 does serve to improve understanding of the life course and roles of the family caregiver. Additionally, on line 77 older age is defined as above 65 years. Adult age is defined on line 78 as 16-44 years. Adding an age range for middle-age individuals is suggested here on line 79. 

Response: We considered ‘middle-aged’ to be 45-64 years old. We have now added the age range for middle-aged individuals as follows (line 78-81):

Line 78-81

“While the family caregivers of individuals with stroke are often in older age rather than middle-aged individuals (45-64years) [22], the family caregivers of individuals with ABI are often middle-aged or older.”

6. Before publication, it may be useful to describe the reason for including the K6 in more detail. Line 171-172 describes its use to validate the LCAS scale, however the K6 only assesses clinically significant serious mental illness. Therefore, an additional explanation of how this validates the LCAS, which includes positive aspects of mental health and the caregiving experience, seems necessary. Line 272 does state that these results are negatively correlated, but again could use clarification earlier. Additionally, the K6 in the English language version is scored from 1-5 where in line 173 the authors state, “each question is scored from 0 to 4”. The 4 point scale anchors described in the text (1 = completely important, 2 = slightly important, 3 = slightly unimportant, 4 =not important at all) are not the same as appear on the English-language version of the instrument (a 7 point scale from “The most deterioration” to “did not change” to “The most improved”) which is a substantially different instrument than the one described.

Response: The K6 assesses clinically significant serious mental illness using the optimal cut-off point, as the reviewer suggested. In addition, a previous study considered the K6 to assess the mental health using a continuous scale, with higher scores reflecting a poorer mental health (Oshio and Kan, 2016). Another previous study further cited a correlation between caregivers’ mental health and the perception of life changes (Negarandeh et al., 2015). Based on the findings from previous studies, we hypothesized that mental health deterioration in cases with higher K6 scores reflected negative life change adaptation with lower LCAS scores. We have now clarified how the K6 validates the LCAS (line 178-184) and corrected the explanation concerning validity in the text (line 280-282). In addition, we apologize for the confusion concerning the 4-point scale for the K6. We used the Japanese version of the K6 developed by Furukawa et al. [35], which is scored from 0-4. We have now added explanations about the 4-point scale for the K6 (line 172-174). 

Line 172-174

“Each question rated the frequency of distress symptoms. The responses were scored as follows: 0 = none of the time, 1 = a little of the time, 2 = some of the time, 3 = a lot of the time, 4 = all of the time; thus, the total score ranges from 0 to 24.”

Line 178-184

“A previous study considered the K6 to be a continuous scale in which higher scores reflect lower levels of mental health [36]. In addition, another study explained that there was a correlation between an adverse impact on caregivers’ mental health and the negative perception regarding life changes [5]. Based on the findings from previous studies, we predicted that mental health deterioration in cases with higher K6 scores reflected negative life change adaptation with lower LCAS scores. Thus, the author used this scale as an indicator of concurrent validity.”

Line 280-282

“A negative life change adaptation as well as a low LCAS score showed a significant amount of variance between mental health deterioration as well as a high K6 score (r=-0.50; P<0.001).”

 

To the comments of Reviewer #2

1. Please re-write the sentences on lines 146-152 so that the procedure is clearer to the reader.

Response: We have carefully corrected (line 147-151).

Line 147-151

“Before sending the survey questionnaires, the author sent informed consent letters to the administrators of all associations of family caregivers of individuals with ABI. Thirty-nine associations (47.5%) consented to participate in the study. The survey questionnaires were distributed to each participant by the staff of each association.” 

2. Table 1 – Change ‘relationship to ABI’ to ‘relationship to individual with ABI’ (so that the terminology is consistent with Table 2.

Response: We have changed “relationship to ABI” to “relationship to individual with ABI” (p.14-15, Table 1).

3. Table 3 is still not legible. Please amend.

Response: We apologize for this difficulty and have now enlarged the letters in Table 3 (p.17-18, Table 3)

4. Table 5 – align the factors to the left of the column

Response: We have now corrected this text (p.20-2l, table 5).

5. The discussion needs to address issues related to the majority of family members are aged 55-74 years and length of time since injury – and how this impacted on the findings regarding adaptation. This does impact on life issues that are impacting on the family. For instance, families who have parents in this age range are more likely to have adult children who are no longer dependent. This is likely to therefore impact on the adaptation of the family to the circumstances they face in relation to ABI.

Response: As suggested, we have now discussed how the family members age and length of the time since injury influenced the findings regarding adaptation (line 314-324).

Line 314-322

“The majority of family members being 55-74 years old and the length of time since the injury being over 10 years were considered to have influenced the findings regarding adaptation. In general, families who have parents in this age range are more likely to have adult children who are no longer dependent. The family caregivers of individuals with ABI must fill the role of caring for an adult who is now dependent after having fulfilled their role of caring for their children. The family caregivers of individuals with ABI therefore need to fill a specific role that most family members never experience. This is likely to influence the adaptation of the family to the circumstances they face in relation to ABI.”

6. Limitations also need to be included

Response: We moved the limitations in response to a previous comment from reviewer #1. Therefore, we have added a new subsection for the limitations to the discussion (line 352).

---

## [Editor Report · Decision Letter 2]

14 Oct 2020

Development of the life change adaptation scale for family caregivers of individuals with acquired brain injury

PONE-D-20-15809R2

Dear Dr. Shindo,

We’re pleased to inform you that your manuscript has been judged scientifically suitable for publication and will be formally accepted for publication once it meets all outstanding technical requirements.

Kind regards,

Manuel Fernández-Alcántara, Ph.D.

Academic Editor

PLOS ONE
---

## [Editor Report · Acceptance letter]

19 Oct 2020

PONE-D-20-15809R2 

Development of the life change adaptation scale for family caregivers of individuals with acquired brain injury 

Dear Dr. Shindo:

I'm pleased to inform you that your manuscript has been deemed suitable for publication in PLOS ONE. Congratulations! Your manuscript is now with our production department. 

Kind regards, 

on behalf of

Dr. Manuel Fernández-Alcántara 

Academic Editor

PLOS ONE